# Intracellular ATP levels in mouse cortical excitatory neurons varies with sleep–wake states

Akiyo Natsubori[1✉], Tomomi Tsunematsu[2,3,4], Akihiro Karashima[5], Hiromi Imamura[6], Naoya Kabe[7], Andrea Trevisiol[8], Johannes Hirrlinger[8,9], Tohru Kodama[1], Tomomi Sanagi[4], Kazuto Masamoto[10], Norio Takata[11], Klaus-Armin Nave[8], Ko Matsui[2], Kenji F. Tanaka[11] & Makoto Honda[1]

Whilst the brain is assumed to exert homeostatic functions to keep the cellular energy status constant under physiological conditions, this has not been experimentally proven. Here, we conducted in vivo optical recordings of intracellular concentration of adenosine 5'-triphosphate (ATP), the major cellular energy metabolite, using a genetically encoded sensor in the mouse brain. We demonstrate that intracellular ATP levels in cortical excitatory neurons fluctuate in a cortex-wide manner depending on the sleep-wake states, correlating with arousal. Interestingly, ATP levels profoundly decreased during rapid eye movement sleep, suggesting a negative energy balance in neurons despite a simultaneous increase in cerebral hemodynamics for energy supply. The reduction in intracellular ATP was also observed in response to local electrical stimulation for neuronal activation, whereas the hemodynamics were simultaneously enhanced. These observations indicate that cerebral energy metabolism may not always meet neuronal energy demands, consequently resulting in physiological fluctuations of intracellular ATP levels in neurons.

[1] Sleep Disorders Project, Tokyo Metropolitan Institute of Medical Science, 2-1-6 Kamikitazawa, Setagaya-Ku, Tokyo 156-8506, Japan. [2] Super-network Brain Physiology, Graduate School of Life Sciences, Tohoku University, Sendai 980-8578, Japan. [3] Precursory Research for Embryonic Science and Technology, Japan Science and Technology Agency, Kawaguchi 332-0012, Japan. [4] Advanced Interdisciplinary Research Division, Frontier Research Institute for Interdisciplinary Sciences, Tohoku University, Sendai 980-8578, Japan. [5] Tohoku Institute of Technology, 35-1, Yagiyama Kasumi-cho, Taihaku-ku, Sendai 982-8577, Japan. [6] Graduate School of Biostudies, Kyoto University, Yoshida-konoe-cho, Sakyo-ku, Kyoto 606-8501, Japan. [7] Neural Prosthesis Project, Tokyo Metropolitan Institute of Medical Science, 2-1-6 Kamikitazawa, Setagaya-ku, Tokyo 156-8506, Japan. [8] Department of Neurogenetics, Max-Planck-Institute for Experimental Medicine, Gottingen 37075, Germany. [9] Carl-Ludwig-Institute for Physiology, University of Leipzig, Liebigstrasse 27, 04103 Leipzig, Germany. [10] Department of Mechanical and Intelligent Systems Engineering, University of Electro-Communications, 1-5-1 Chofugaoka, Chofu, Tokyo 182-8585, Japan. [11] Department of Neuropsychiatry, Keio University School of Medicine, 35 Shinanomachi, Shinjuku-ku, Tokyo 160-8582, Japan. ✉email: natsubori-ak@igakuken.or.jp

Energy homeostasis is crucial for enabling vital cellular activities. In the brain, such homeostatic mechanisms are often described as "neurometabolic coupling" (NMC) between high-frequency neuronal oscillations that are involved in energy expenditure and hemodynamics or glucose metabolism[1–4]. These couplings have been widely used for functional brain imaging such as functional magnetic resonance imaging (fMRI) and positron emission tomography (PET) and could locally function to maintain a constant cellular energy status.

Furthermore, global brain energy homeostasis could also function in a state-dependent manner. Across the sleep-wake states of animals, brain metabolic activities for energy supply such as hemodynamics and glucose metabolism and multiple cellular energy-consuming activities including neuronal firings simultaneously fluctuate over wide brain areas[5–8]. In the wake state, brain-wide metabolic activities and neuronal activities increased, and in the non-REM sleep state, they decreased[5–8]. These global parallel dynamics of energy producing and consuming activities across the sleep-wake states could be brought about by the brain monoaminergic systems[5,9]. Under these local and global brain energy homeostatic mechanisms, it is hypothesized that the cellular energy status in the brain could be maintained constant across the sleep-wake states of animals and cellular energy depletion could be prevented in all physiological conditions. However, this has not been experimentally proven.

Adenosine 5′-triphosphate (ATP), the major cellular energy metabolite, is synthesized to meet requirements for many sub-cellular processes in the brain such as synaptic transmissions, action potentials, and maintaining resting membrane potentials[10]. Based on ex vivo studies, intracellular ATP levels in neurons decrease following high-frequency electrical stimulation and glutamate exposure, as well as in response to pharmacological inhibition of ATP production[11–13]. Thus, intracellular ATP concentrations can be regarded as the cellular energy status reflecting both energy supply and consumption. Nevertheless, the currently known mechanisms regulating intracellular ATP levels in neurons are based on ex vivo observations and may, therefore, differ in vivo, as ex vivo conditions do not reproduce local or global brain energy homeostatic mechanisms.

To investigate whether the cellular energy status is maintained constant in the brain of living animals, we conducted in vivo optical measurements of intracellular ATP dynamics in neurons using a genetically encoded fluorescent ATP sensor[11,14]. With fiber photometric recordings, we observed that in vivo intracellular ATP levels in cortical neurons decreased under local electrical stimulation, which was compatible with the results of a previous ex vivo report[11], whereas hemodynamic responses were also induced in terms of local brain energy homeostatic functions. Furthermore, fiber photometric recordings and wide-field microscopic imaging revealed that global fluctuations in intracellular ATP levels of cortical neurons depend on the sleep-wake states of animals, presumably affected by local and global brain energy homeostatic mechanisms. Simultaneous in vivo recording of neuronal activity and cerebral hemodynamics could help us understand the unique characteristics of intracellular ATP dynamics in neurons and mechanisms of brain energy homeostasis as a whole.

## Results

### Electrical stimulation diminishes ATP levels in neurons.
It was previously shown that ex vivo, high-frequency electrical stimulation led to a decrease in axonal ATP levels, possibly reflecting elevated ATP consumption[11]. However, these observations may not reflect the in vivo conditions, since brain metabolic activities related to energy supply, including blood flow, were also shown to

focally enhance following local electrical stimulation[15,16]. To evaluate the effect of local brain energy homeostatic functions to maintain the neuronal energy status constant, we observed in vivo neuronal intracellular ATP responses under local electrical stimulation, compatible with previous ex vivo observations[11].

To examine ATP dynamics in response to electrical stimulation in vivo, we monitored intracellular ATP levels in layers 5 and 6 of the cerebral cortex (primary motor area) using Thy1-ATeam transgenic mice, in which the genetically encoded ATP indicator, AT1.03[YEMK] (ATeam), was expressed in pyramidal neurons under the control of a Thy1 promoter[11] (Fig. 1a). In this mouse line, we recorded the compound intracellular ATP signals, with a fiber photometric system[17–19], as the ratio of fluorescence intensity of yellow fluorescent protein (YFP) to cyan fluorescent protein (CFP) (Y/C ratio), which was based on a Förster resonance energy transfer (FRET)[14] occurring on a timescale of microseconds. As changes in pH can modulate fluorescence from some versions of YFP[20], we inspected the pH dependency of the FRET signal of purified AT1.03[YEMK] constructs[14]. The fluorescence emission ratio was almost invariant at pH conditions >7.3 (Supplementary Fig. 1). The ratio slightly decreased as the decrease of cellular pH lower than 7.3 when the cytosolic ATP levels were higher than 2.0 mM. These findings suggest that the FRET signal from AT1.03[YEMK] would not be heavily corrupted by cellular pH fluctuations[21–23].

During cortical electrical stimulation with the electrode adjacent to the optical fiber (Fig. 1b), intracellular ATP levels significantly decreased at frequencies of 10, 30, 50, and 100 Hz in a stimulation frequency-dependent manner ($p < 0.0001$, $F = 16.99$, one-way analysis of variance (ANOVA); $n = 6$ mice; Fig. 1c, d). Note that our electrical stimulation protocols (stimulation intensity: 100 μA and frequencies: 1-100 Hz) were adopted in reference to in vivo electrical stimulation protocols for inducing neurogenic vascular responses in previous studies[15,16], whereas stimulation with higher frequencies would induce the neuronal firings whose patterns were beyond the spontaneous ones[24]. To investigate the dynamics of neuronal activity and accompanying cerebral metabolic activity evoked by the electrical stimulation, we monitored intracellular $Ca^{2+}$ activities in pyramidal neurons using fiber photometry and measured cerebral blood flow (CBF) using laser Doppler flowmetry (LDF) (Fig. 1e-h). To monitor the $Ca^{2+}$ activities in pyramidal neurons, we injected a virus that expresses CaMKII-GCaMP (AAV9-CaMKIIa-jGCaMP7f) to layer 5 of the cerebral cortex (primary motor area). The CaMKII-GCaMP signals were enhanced by local electrical stimulation depending on the stimulation frequency ($p < 0.0001$, $F = 44.91$, one-way ANOVA, $n = 6$ mice; Fig. 1e, f), which is consistent with the results of a previous report[25]. A similar enhancing effect was found regarding CBF ($p = 0.0003$, $F = 7.21$, one-way ANOVA, $n = 7$ mice; Fig. 1g, h). The maximum amplitude of the ATP signal significantly decayed to the end of stimulation ($p < 0.001$ vs. time of the end of stimulation; Student's $t$-test with Bonferroni correction) and occurred significantly later compared with those of CaMKII-GCaMP and CBF signals at frequencies of 30, 50, and 100 Hz ($p < 0.01$ vs. CaMKII-GCaMP and CBF; Student's $t$-test with Bonferroni correction; Fig. 1i). The ATP signal also showed a significantly longer recovery half-time compared with CaMKII-GCaMP or CBF signals at frequencies of 10, 30, 50, and 100 Hz ($p < 0.05$ vs. CaMKII-GCaMP and CBF; Student's $t$-test with Bonferroni correction; Fig. 1j). These results suggest that local electrical stimulation induce long-lasting, negative effects on the energy balance within cortical neurons in a stimulation frequency-dependent manner, despite a simultaneous enhancement in CBF. These findings suggest that local brain energy homeostatic functions, including hemodynamics, could not be sufficient to always maintain the neuronal ATP levels constant under the

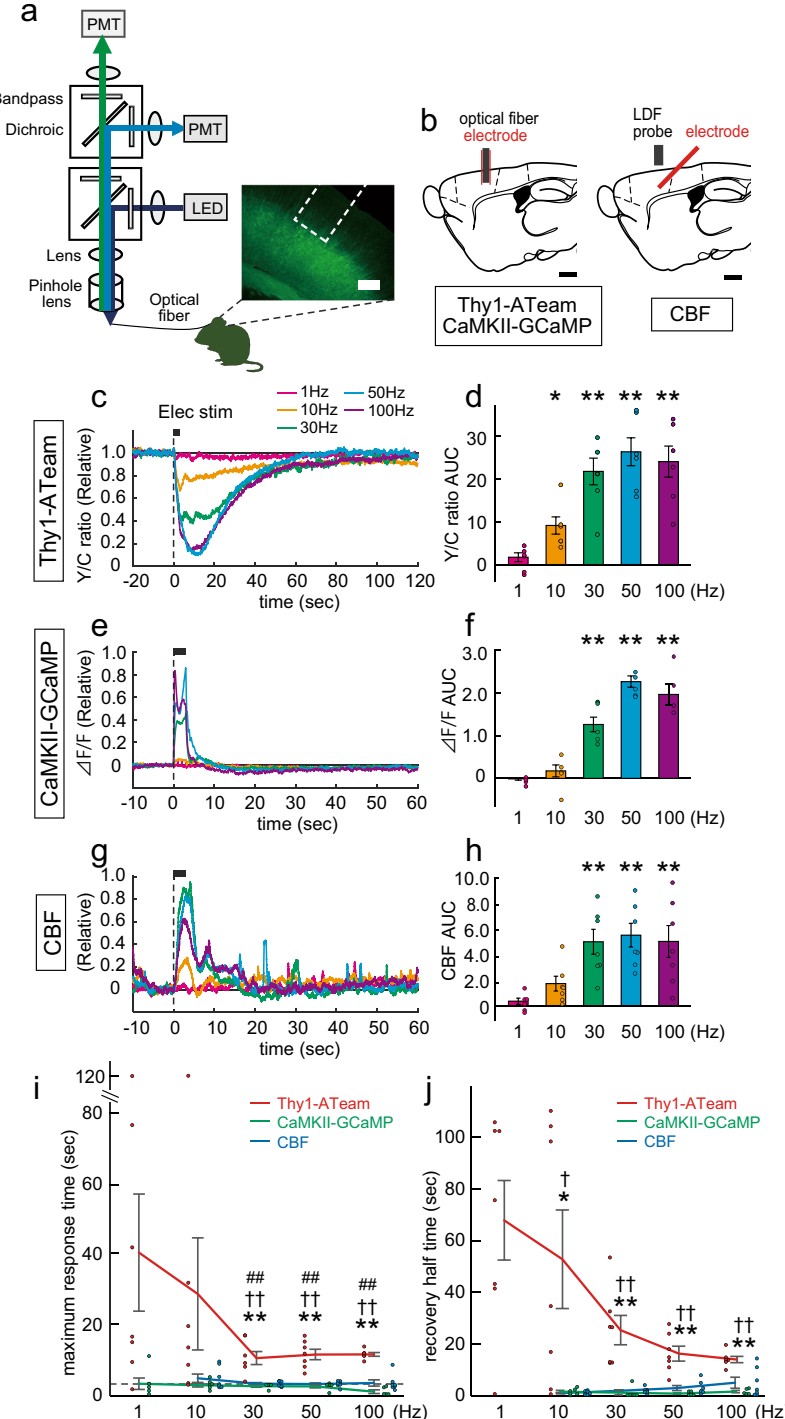

electrical stimulation but could slowly complement stimulation-induced neuronal energy consumption. Considering the classical use of in vivo electrical stimulation methodologies[26,27], our observation of electrical stimulation-induced long-lasting neuronal ATP reduction could be of interest.

**State-dependent variation of neuronal cytosolic ATP levels.** We next investigated to what extent intracellular ATP levels within the cortical neurons of live animals may physiologically fluctuate depending on the state they are in using the fiber photometric system. We observed that the ATP signals (Y/C ratio) in the cortex of Thy1-ATeam mice showed slow fluctuations, with frequencies predominantly lower than 0.05 Hz and with durations of

convex waves of 30-60 s (Fig. 2a). No waves of the Y/C ratio were observed in control (wildtype) mice. To characterize the intracellular ATP dynamics within cortical neurons, we next monitored ATP signals across the sleep-wake cycles of animals. We simultaneously recorded the CBF using LDF as a metabolic parameter for energy supply, which is controlled by local neuronal activities[28] (Fig. 2b). We observed that ATP levels fluctuated across the sleep-wake cycles and negatively correlated with CBF (Fig. 2c). Intriguingly, ATP levels significantly decreased during rapid eye movement (REM) sleep state compared with wake and non-REM (NREM) sleep state (main effect for state using one-way ANOVA: $F(2, 9) = 21.35$, $p < 0.001$; followed by multiple comparisons with the Bonferroni test: $p < 0.05$ for REM

**Fig. 1 Local electrical stimulation evokes intracellular ATP response in cortical neurons in vivo. a** Schematic illustration of the fiber photometric system. For the recording of Thy1-ATeam mice, the fluorescence emission is separated by a dichroic mirror. Yellow and cyan fluorescence signals are corrected by band-pass filters and enhanced by photomultiplier tubes (PMTs). For the recording of CaMKII-GCaMP, one light pathway for fluorescence emission and PMT was used. Dashed white line, optical fiber. LED, light-emitting diode. Scale bar = 200 μm. **b** Schematic illustration of optical fiber with a stimulation electrode implanted in the cortex for Thy1-ATeam or CaMKII-GCaMP monitoring. To monitor cerebral blood flow (CBF), the laser Doppler flowmetry (LDF) probe is positioned over the skull where the electrode is implanted. Scale bar = 1 mm. **c** ATP signals during local electrical stimulation at different frequencies. Signals are normalized between one (mean of 5 s before stimulation) and zero (trough value after 50 Hz stimulation) in each animal and presented as mean ($n = 6$ mice). **d** ATP signals represented as area under the curve (AUC). $*p < 0.05$ and $**p < 0.01$; one-way ANOVA, Bonferroni test (AUC value for 5 s before stimulation vs. that for 60 s after stimulation; $n = 6$ mice), mean ± SEM. **e** CaMKII-GCaMP signals during local stimulation. Signals are normalized between zero (mean of 5 s before stimulation) and one (peak value after 50 Hz stimulation) in each animal and presented as mean value ($n = 6$ mice). **f** The AUC of CaMKII-GCaMP signals following stimulation. **g** CBF during local stimulation. Normalization as in (**e**) ($n = 7$ mice). **h** CBF response as AUC. **i** Maximum response time of Thy1-ATeam, CaMKII-GCaMP, and CBF following stimulation. $**p < 0.01$ vs. CaMKII-GCaMP; $††p < 0.01$ vs. CBF; $##p < 0.01$ vs. time of the end of stimulation; Student's $t$-test with Bonferroni correction, mean ± SEM. **j** Recovery half-time of Thy1-ATeam, CaMKII-GCaMP, and CBF following stimulation. $*p < 0.05$ and $**p < 0.01$ vs. CaMKII-GCaMP; $†p < 0.05$ and $††p < 0.01$ vs. CBF; Student's $t$-test with Bonferroni correction.

sleep vs. wake and NREM sleep, respectively, $n = 4$ mice; Fig. 2d). In contrast, CBF increased during REM sleep state ($F(2, 9) = 21.63$, $p < 0.001$ and $p < 0.05$ for REM sleep vs. wake and NREM sleep, respectively, $n = 4$, the same mice; Fig. 2e), which, in combination with a similar increase in other metabolic activities related to energy supply in this state[6–8,29], suggests a negative energy balance in cortical neurons and reflects a high energy expenditure during REM sleep state.

Focusing on changes in ATP signals and CBF during sleep state transitions, ATP levels decreased following the transition from the wake state to NREM sleep state, whereas CBF transiently decreased during transition but was restored afterward (main effect for state using two-way ANOVA: $F(11, 33) = 4.73$, $p < 0.001$ for ATP and $F(11, 33) = 5.87$, $p < 0.001$ for CBF; followed by multiple comparisons with the Bonferroni test: $p < 0.05$ vs. the data of the first and fourth epochs; $n = 4$ mice; Fig. 2f(i). During transitions from NREM sleep state to wake state, ATP signals increased, whereas CBF transiently increased and then significantly decreased ($F(11, 33) = 5.78$, $p < 0.001$ for ATP and $F(11, 33) = 8.24$, $p < 0.001$ for CBF; followed by multiple comparisons with the Bonferroni test: $p < 0.05$ vs. the data of the first and fourth epochs; $n = 4$ mice; Fig. 2f(ii). When transitioning from NREM to REM sleep, ATP signals gradually decreased, whereas CBF increased ($F(11, 33) = 10.62$, $p < 0.001$ for ATP and $F(11, 33) = 5.13$, $p < 0.001$ for CBF; followed by multiple comparisons with the Bonferroni test: $p < 0.05$ vs. the data of the first and fourth epochs; $n = 4$ mice; Fig. 2f(iii). In contrast, at the transition from REM sleep state to the wake state, ATP signals promptly increased, whereas CBF gradually decreased ($F(11, 33) = 9.63$, $p < 0.001$ for ATP and $F(11, 33) = 10.56$, $p < 0.001$ for CBF; followed by multiple comparisons with the Bonferroni test: $p < 0.05$ vs. the data of the first and fourth epochs; $n = 4$ mice; Fig. 2f(iv). These changes were accompanied by changes in electroencephalographic (EEG) parameters (delta power and theta frequency) and electromyographic (EMG) activity, which match the defining features of the sleep and wake states.

As the intracellular ATP levels in cortical neurons decreased during the transitions from the wake state to NREM sleep state and from NREM to REM sleep states, we assumed that they may be dependent on the depth of NREM sleep and/or sub-state of REM sleep between the tonic and phasic components. However, the temporal cross-correlation analysis revealed no obvious correlation between ATP signals and EEG delta power during NREM sleep state or theta frequency during REM sleep state[30,31] (Supplementary Fig. 2). These data indicate that the intracellular ATP levels of cortical neurons do not correlate with the depth of NREM sleep or sub-state of REM sleep.

To further assess intracellular ATP dynamics in cortical neurons in response to different states, we used isoflurane-mediated anesthesia, which triggers a state of unconsciousness that shares striking similarities with those during NREM sleep, regarding the activities in the cerebral cortex and involved regions[32,33]. Isoflurane treatment reversibly decreased intracellular ATP levels in cortical neurons ($F(2, 8) = 21.24$, $p < 0.001$, main effect for isoflurane using two-way ANOVA; followed by multiple comparisons with the Bonferroni test, $p < 0.05$; pre vs. iso and iso vs. post; $n = 5$ mice; Supplementary Fig. 3), which was accompanied by alterations in EEG and EMG activities. This reduction in ATP levels was comparable with those during REM sleep and following electrical stimulation at 50 Hz (Supplementary Fig. 3B). General anesthesia agents, including isoflurane, diminish global neuronal and glial activities in the cortex[31,34]. These suggest that isoflurane-mediated anesthesia results in a negative energy balance in neurons, which could reflect the suppression of ATP production related to energy supply that exceeds the reduction in neuronal activity and, therefore, energy consumption[35,36].

**Rapid changes in neuronal ATP levels between sub-states.** Given our finding that intracellular ATP levels of cortical neurons show fluctuations within the sleep-wake states and at transitions between these states (Supplementary Fig. 4), we next examined these fluctuations in more detail. During the wake state, two distinct sub-states are defined: quiet-awake (non-movement) and active-awake (locomotion or other movements)[37]. As these sub-states are associated with distinct neuronal activities in the cortex[37], intracellular ATP levels within these cells may also vary between sub-states. To address this possibility, we evaluated neuronal ATP dynamics at the transition from quiet- to active-awake, which is characterized by the onset of EMG activity. ATP signals increased and peaked within a few seconds after the transition from quiet- to active-awake, and remained elevated thereafter (Fig. 3a, b). CBF immediately increased at the transition, reaching its highest level preceding the peak in ATP concentrations, and following a transient drop resumed a level almost below that during quiet-awake. The transition between sub-states was confirmed by EEG gamma power[37], the increase of which was detectable prior to the enhancement of CBF and EMG activities (Fig. 3a).

As we also observed fluctuations in ATP signals and CBF during NREM sleep state (Supplementary Fig. 4), we next assessed these parameters at micro-awakenings during NREM sleep state[38]. ATP signals transiently increased following a brief elevation in EEG gamma power and short-term EMG activity (Fig. 3c, d). During ATP increase, CBF increased concomitantly with the rise in EMG activity, followed by a transient decrease

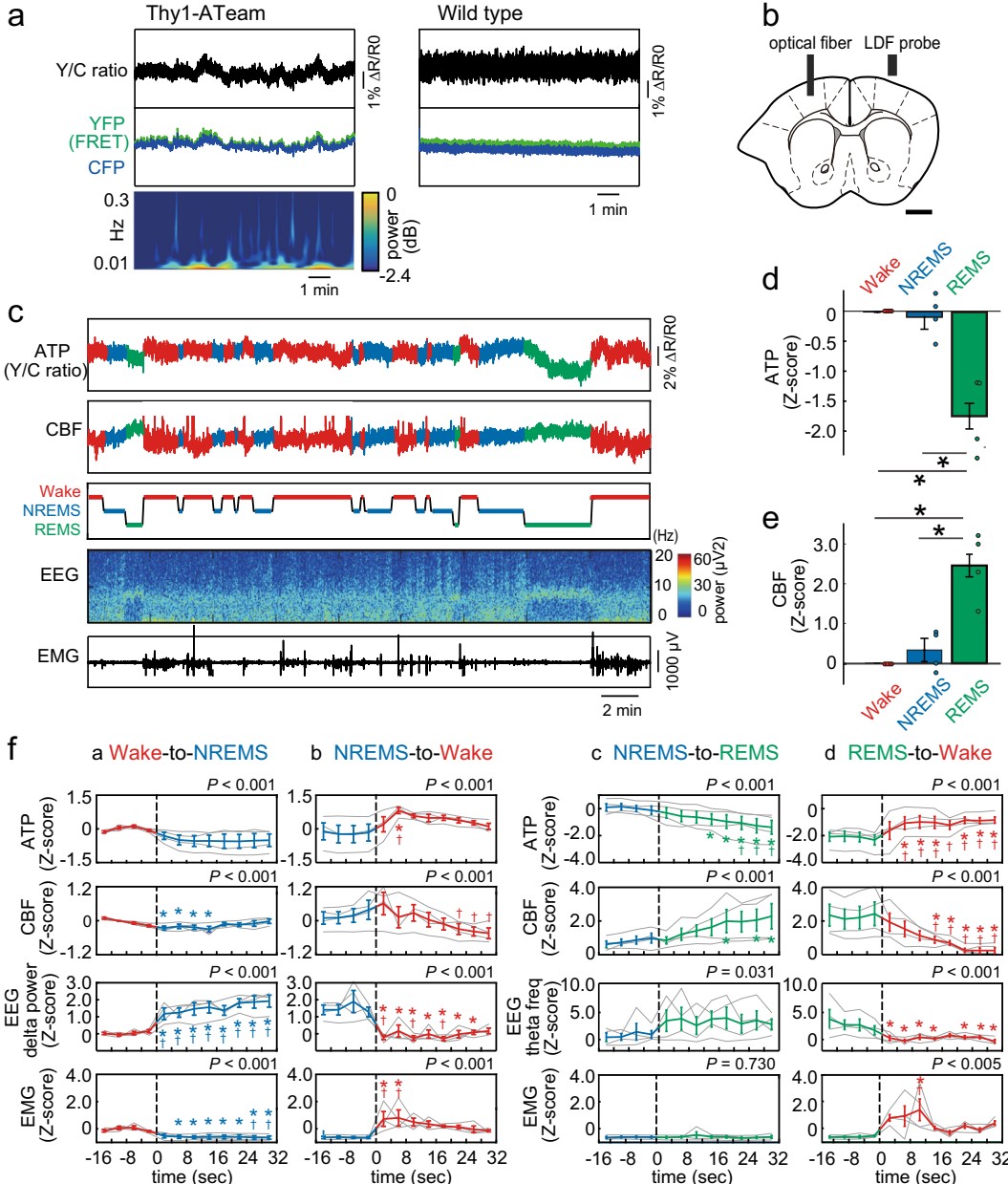

**Fig. 2 Dynamics of intracellular ATP in cortical neurons and CBF across the sleep-wake states. a** Example of the yellow/cyan fluorescence intensity ratio (Y/C ratio) and corresponding intensity changes (YFP and CFP) recorded in the cortex in Thy1-ATeam and wildtype control mice. In Thy1-ATeam mice, corresponding power spectral data are attached. **b** Schematic illustration of simultaneous recording of cortical ATP signals through an optical fiber and of cerebral blood flow (CBF) using a laser Doppler flowmetry (LDF) whose probe is positioned above the skull over the contralateral cortical hemisphere. Scale bar = 1 mm. **c** Representative traces of intracellular ATP signals (Y/C ratio) in cortical neurons, CBF, vigilance states (wake, non-REM sleep (NREMS), and REM sleep (REMS)), EEG power density spectrum, and EMG. **d** Mean intracellular ATP levels during the wake, NREMS, and REMS states. *$p < 0.05$, two-way ANOVA followed by the Bonferroni post-hoc test ($n = 4$ mice), mean ± SEM. (**e**) Mean CBF values during the wake, NREMS, and REMS states ($n = 4$ mice, same animals as in (**d**)), mean ± SEM. (**f**) ATP signals, CBF, delta power or theta frequency, and EMG activity for the transitions of Wake-to-NREMS (i), NREMS-to-Wake (ii), NREMS-to-REMS (iii), and REMS-to-Wake (iv). Transitions occurred at 0 s. Data are from 4-s intervals characterized by state transitions. Values are averaged across all such transitions exhibited by each animal. Group means ± SEMs of these averaged values are shown ($n = 4$ mice). *P*-values denote significant differences between the states (two-way ANOVA with the Bonferroni post-hoc test). *$p < 0.05$ vs. first epoch before state transition and †$p < 0.05$ vs. fourth epoch immediately before state transition.

below the baseline before its level normalized. Interestingly, unlike CBF, ATP levels did not drop below the baseline following the increase, suggesting that prompt adjustment of metabolic activities enabled timely energy supply and a positive energy balance in neurons.

These temporal relationships between the dynamics of ATP levels and CBF during the wake and NREM sleep states were

confirmed by temporal cross-correlation analysis (Supplementary Fig. 5). ATP signals showed a positive correlation peak with a positive time lag (a few seconds) in the wake and NREM sleep states, and a negative correlation peak with a negative time lag (a few seconds) to CBF in the wake state. This biphasic correlation between the ATP levels and CBF was not observed in REM sleep state. These data support our findings that CBF increased

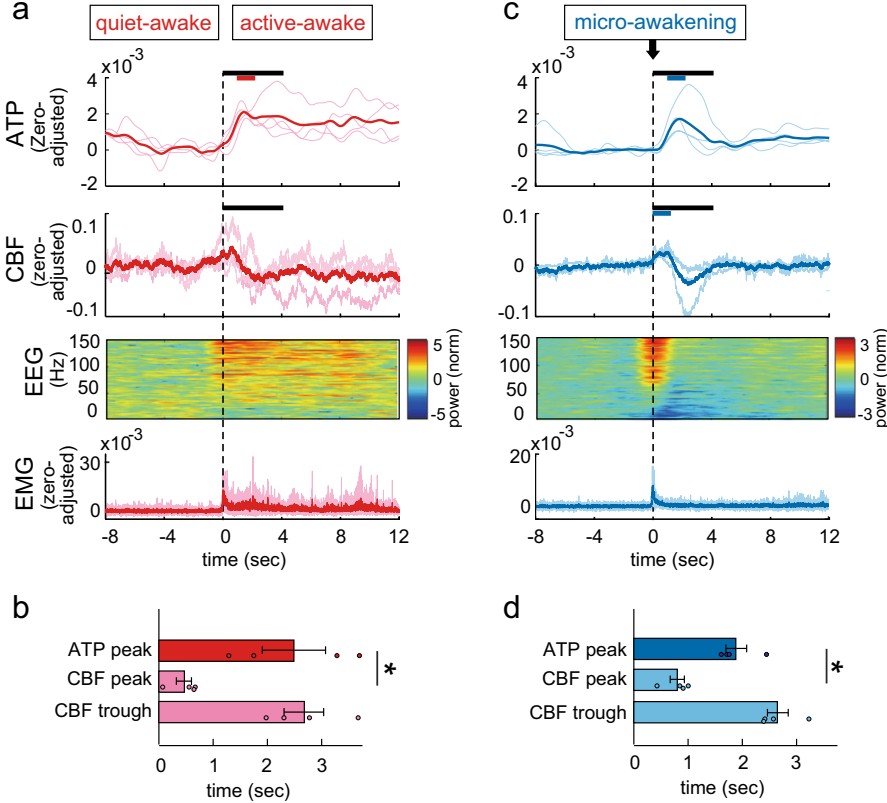

**Fig. 3 Dynamics of intracellular ATP in cortical neurons and CBF during sub-state changes. a** Trace of averaged ATP and CBF signals, and EEG power ($n = 4$ mice) through the transition from quiet-awake to active-awake states. Transitions occurred at 0 s. Black bar = statistical period. Colored bar = $p <$ 0.05; paired $t$-test with Bonferroni correction (during 4 s just before the onset of active-awake vs. that during each 1 s period for 8 s after the onset of active-awake, $n = 4$ mice). (**b**) Peak time of ATP signal and peak and trough time of CBF signal after the onset of active-awake. *$p < 0.05$; Student's $t$-test with Bonferroni correction ($n = 4$ mice), mean ± SEM. **c** Trace of averaged ATP and CBF signals and EEG power ($n = 4$ mice) at micro-awakening during the non-REM sleep state. Micro-awakening occurred at 0 s. **d** Peak time of ATP signal and peak and trough time of CBF signal after the onset of micro-awakening. *$p < 0.05$, **$p < 0.01$; Student's $t$-test with Bonferroni correction ($n = 4$ mice), mean ± SEM.

preceding the peak in ATP levels and following a transient drop in the wake and NREM sleep states (Fig. 3).

**Cortical synchronization of neuronal cytosolic ATP dynamics**. Since neuronal activity within the cortex fluctuates across the sleep-wake states locally as well as globally[39,40], we wondered whether the observed dynamics in ATP levels between these states is cortex-wide or region-specific. We monitored the intracellular ATP dynamics over a broad area of the cerebral cortex using wide-field imaging in Thy1-ATeam mice (Fig. 4, Supplementary Movie 1-3). Remarkably, intracellular ATP levels changed across the sleep-wake states throughout the monitored area and significantly decreased during REM sleep state (Fig. 4a, b and Supplementary Fig. 6) (main effect for state using one-way ANOVA: $F(2, 12) = 81.04$, $p < 0.001$; followed by multiple comparisons with the Bonferroni test: $p < 0.05$, REM vs. wake and NREM sleep, $n = 5$ mice; Fig. 4b), which is consistent with the above-described observations. To analyze the spatio-temporal ATP dynamics, we used the six cortical modules as regions of interest (ROIs) (Fig. 4c). As expected from the findings above, state-dependent ATP dynamics were observed in all ROIs (main effect for state using two-way ANOVA: $F(2, 9) = 880.3$, $p < 0.001$; followed by multiple comparisons with the Bonferroni test: $p < 0.05$, REM sleep vs. wake and NREM sleep, $n = 6$ ROIs; Fig. 4d). At the transitions between states, intracellular ATP levels in cortical neurons synchronously changed in all ROIs (Fig. 4e, Supplementary Movie 1-3). To elucidate the characteristics of synchronization in cortical ATP dynamics, we next assessed the

correlation in ATP dynamics between the ROIs across and within the states. Across the wake, NREM sleep, and REM sleep states, the ATP dynamics were highly synchronized among all cortical regions (Fig. 4f). Within each state, the ATP dynamics showed the highest correlation in symmetric ROIs between the right and left hemispheres (Fig. 4g). In one cerebral hemisphere, the cortical ATP dynamics showed higher correlations in the adjacent ROIs compared with non-adjacent ROIs. These characteristics of synchronization in cortical ATP dynamics were consistently observed during the wake, NREM sleep, and REM sleep states, and their averaged correlation intensity among ROIs was not significantly altered across the three states (main effect for state using one-way ANOVA: $F(2, 9) = 0.86$, $p = 0.46$, $n = 4$ mice; Fig. 4h).

**Discussion**
The present study showed that intracellular ATP levels in cortical excitatory neurons fluctuate under physiological conditions depending on the sleep-wake states of animals. Neuronal ATP levels increased simultaneously with the level of the arousal of animals, whereas they profoundly decreased in an REM sleep-specific manner. These in vivo observations were achieved by genetically encoded ATP sensors recently developed and adopted[12,14,23,41–47], as well as fiber photometry and wide-field microscopy, which was originally used for in vivo monitoring of neuronal calcium activities[19,48]. Our data demonstrate the superiority of these techniques over previously used, conventional methodologies such as chemiluminescence imaging by

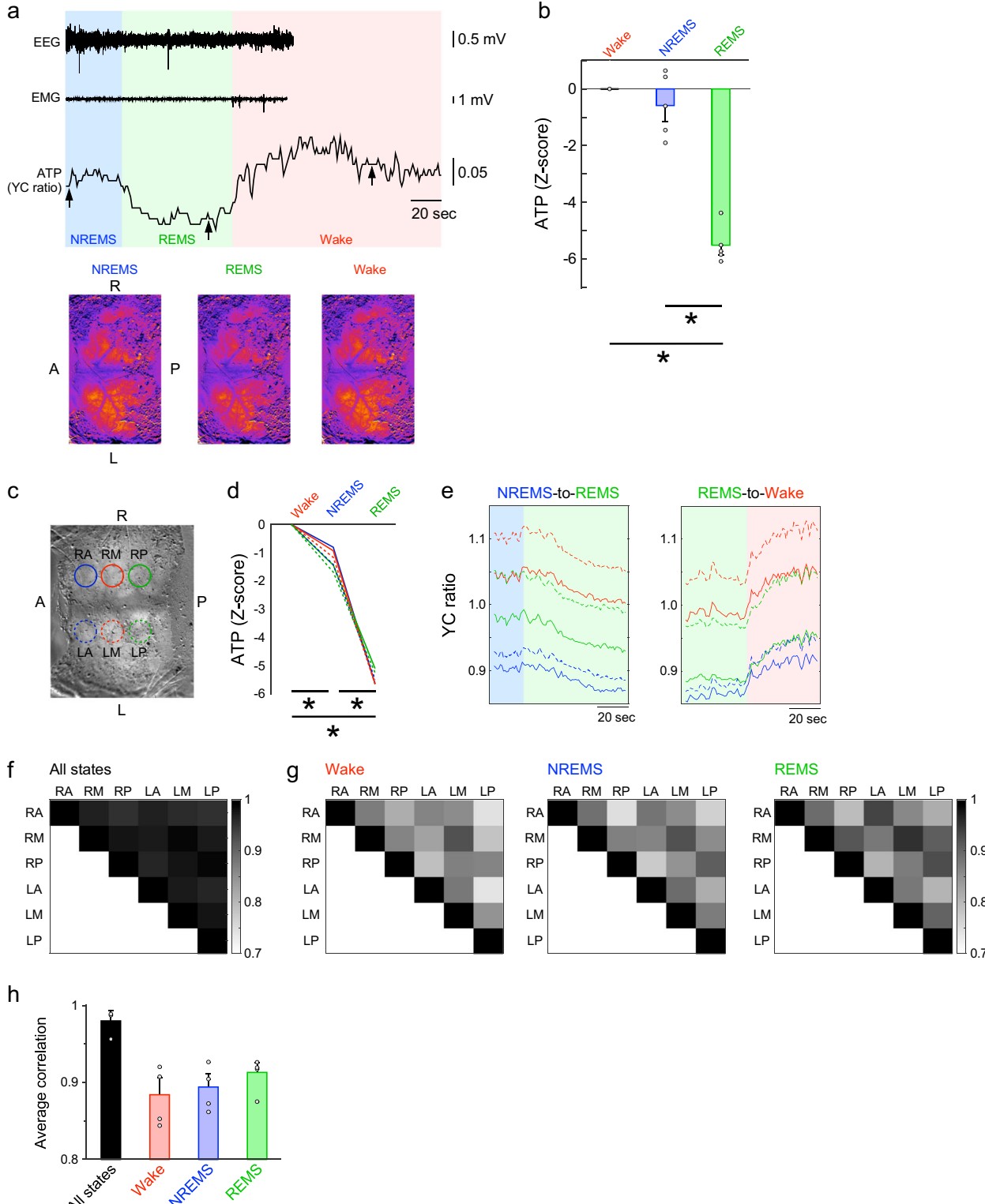

luciferase[49], which lack temporal and spatial resolutions, are sensitive to a high pH, and importantly, are restricted to in vitro/ex vivo application.

Our most surprising finding was that intracellular ATP levels in cortical excitatory neurons decrease during REM sleep state (Fig. 2). Even though this sleep state is known to be crucial in animals including humans, cortical neuronal activities related to REM sleep and corresponding functions have not been elucidated. The reduction in ATP concentrations in cortical excitatory neurons during REM sleep state indicates a negative energy balance in these cells, whereas the CBF and glucose metabolism for energy supply were simultaneously enhanced[6–8,29]. These observations suggest an REM sleep-specific enhancement of energy-consuming activities and/or disturbance of ATP production in cortical neurons. A rise in energy consumption related to neuronal firing activity seems unlikely, as this was reported to be decreased in cortical pyramidal neurons during REM sleep state[48,50]. As another characteristic of metabolic activity during

**Fig. 4 Wide-field optical imaging of intracellular ATP dynamics in cortical neurons. a** Representative ATP signals, with EEG and EMG signals, in the whole cortical area across the sleep-wake states imaged in one example mouse. An imaging window was placed over the cortex. Arrows indicate corresponding time points of imaging at the bottom. A: anterior, P: posterior, R: right, L: left. Scale bar = 1 cm. **b** Mean ATP signals in the observed cortical area during the wake, non-REM sleep (NREMS), and REM sleep (REMS). *$p < 0.05$, one-way ANOVA with the Bonferroni post-hoc test. Data are presented as mean ± SEM ($n = 5$ mice). **c** Circles depict the six ROIs in which ATP signals were analyzed. RA: right-anterior, RM: right-middle, RP: right-posterior, LA: left-anterior, LM: left-middle, LP: left-posterior. Scale bar = 1 cm. **d** Mean ATP signals during the wake, NREMS, and REMS states in six ROIs from one example mouse. Colors correspond to ROI circles in (**c**). *$p < 0.05$, two-way ANOVA with the Bonferroni post-hoc test ($n = 6$ ROIs). **e** ATP signals in six ROIs from one animal for the transitions of NREMS-to-REMS and REMS-to-wake. Transitions occurred at 0 s. Data are from 4-s intervals characterized by state transitions. **f** Correlation map between six ROIs across the sleep-wake states. The data were averaged from four mice. **g** Correlation maps between six ROIs in the wake, NREMS, and REMS states. Note that higher correlations were observed between hemisphere-symmetric ROIs (RA-LA, RM-LM, and RP-LP) and neighboring ROIs (e.g., RA-RM or LM-LP). Data are averaged from four mice. **h** The comparison of correlation coefficients among states. Data are presented as mean ± SEM ($n = 4$ mice). See also Supplementary Movie 1-3 and Supplementary Fig. 6.

REM sleep state, high levels of extracellular lactate were reported[51], suggesting that anaerobic glycolysis could be promoted in neurons and/or astrocytes during REM sleep state, whose predominance should be the cause of inefficient ATP production. Furthermore, it is assumed that metabolic processes involved in ATP production such as oxidative phosphorylation may markedly decelerate during REM sleep state. In the last step of oxidative phosphorylation, brain thermogenesis likely occurs via the activation of uncoupling proteins, which could block ATP synthesis[52,53]. Interestingly, previous studies have reported an increase in brain temperature during REM sleep state[54–56], thus supporting our hypothesis that active metabolic heat production could cause the reduction in ATP levels during this state[5]. Our technique and data presented may help unravel the importance of REM sleep and its implications for the brain in future studies.

We found that intracellular ATP levels in cortical neurons increased at the onset of the wake state upon transition from quiet- to active awake, as well as following micro-awakening during NREM sleep state, and in all cases was preceded by CBF increase (Figs. 2, 3). These findings indicate a correlation between intracellular ATP concentrations and the level of arousal of animals caused by a rise in energy production in response to increased activity. Focusing on ATP dynamics within the different states, a transient peak in ATP signals was observed a few seconds following the transition of quiet- to active-awake and after the event of micro-awakening, subsequent to an increase in EEG gamma power and CBF (Fig. 3). The coupling between cortical gamma oscillation and induced CBF enhancement via glutamate-mediated synaptic transmission is known as NMC[1–4,28]. This mechanism may explain the transient increase in intracellular ATP levels in cortical neurons, EEG gamma activity, and CBF, by allowing a transient excess in energy supply, and therefore positive energy balance in neurons during the wake and NREM sleep states. Interestingly, a time lag of 1-2 s in the ATP peak after the CBF peak (Fig. 3) was consistent with a time lag in the increase of tissue $PO_2$ after the CBF increase, which could be caused by the time required for oxygen to diffuse through the tissue for ATP production[57].

Our wide-field imaging further revealed that the intracellular ATP dynamics in neurons showed cortical synchronization across the sleep-wake states (Fig. 4), reflecting cortex-wide neuronal activities and hemodynamics[48,58]. It is imaginable that these state-dependent cortex-wide fluctuations of cellular energy status could be affected by volume transmission of neuromodulators including of norepinephrine and acetylcholine, as these were shown to be released in a cortex-wide manner depending on the arousal state of animals[59,60]. Furthermore, the signals of neuromodulators regulate cerebral metabolism as well as neuronal and glial cellular activities[61–65]. A general cortex-wide synchronization in ATP dynamics was also found within each state. In more detail, however, a relatively higher correlation was observed

between symmetrical cortical regions and neighboring regions, whereas correlations were slightly lower between ROIs further away from each other. This region-specificity of cortical ATP dynamics may be related to the functional connectivity of networks and the underlying neuronal activities. In this case, it would be intriguing to test whether ATP dynamics would be enhanced/changed in their regional specificity in response to certain behavioral activities or sensory stimulation of animals.

Neurons express ATP-sensitive potassium channels ($K_{ATP}$ channels) that are part of the machinery that enables electrical excitability[66]. However, since little is known about in vivo intracellular ATP dynamics in neurons, the specific functions and characteristics of these channels under in vivo physiological as well as pathological conditions have not been fully elucidated. We discovered sleep-wake state-dependent fluctuations of intracellular ATP levels in neurons, which strongly supports the assumption that such levels directly regulate neuronal firing activity via $K_{ATP}$ channels under physiological conditions. Furthermore, the physiological intracellular ATP dynamics we observed provides strong supporting evidence to previous reports regarding the effect of brain energy metabolism on neuronal activity, and ultimately, on a myriad of brain functions such as memory formation and maintenance[67].

Despite the power of the experimental setup used in this study, it still has technical limitations, namely the difficulty in calibrating the ATP-sensor signal to absolute cytosolic concentrations in vivo. Therefore, further technical advances will be needed for more precise measurements of the absolute concentration of neuronal intracellular ATP. As a consequence, we currently do not know whether the in vivo intracellular ATP concentration in neurons is the same as that ex vivo, which has been estimated at approximately 2 mM[21,23,45]. Nevertheless, the here measured ATP levels following electrical stimulation, which are within the dynamic range of the ATeam sensor with its dissociation constant ($K_d$) of 1.2 mM[14], are likely to be comparable with the ATP response found ex vivo in this context using ATeam[11].

In order to examine brain activities and functions, neuronal electrical activities and coupled brain metabolic activities (via fMRI and PET imaging) have been utilized. In contrast, the in vivo intracellular ATP levels in cortical neurons observed in this study showed distinct dynamics across the sleep-wake states that differ from those found using the above conventional methodologies. We also observed that intracellular ATP levels decreased upon local electrical stimulation, triggering a rise in both energy supply and consumption, whereas general anesthesia had a similar effect on ATP levels but simultaneously inhibited energy supply and consumption[32,34–36] (Fig. 1 and Supplementary Fig. 2). Taken together, the intracellular ATP levels, which could be regarded as the cellular pool of energy, can be useful for understanding physiological and pathological processes in the brain as well as in the body of live animals, such as hibernation,

fatigue, metabolic disorders including ischemia, and degenerative diseases.

## Methods

**Animals**. All animal procedures were conducted in accordance with the National Institutes of Health Guide for the Care and Use of Laboratory Animals and were approved by the Animal Research Committee of Tokyo Metropolitan Institute of Medical Science (approval No. 16017) and Tohoku University (approval No. 2018LsLMO-020). All efforts were made to minimize animal suffering or discomfort and to reduce the number of animals used. Experiments were performed using 3- to 12-month-old male and female mice. Thy1-ATeam transgenic mice[11] were used for ATP recording and wildtype mice were used for calcium or CBF recordings under the electrical stimulation. The genetic background of all transgenic and wildtype mice was C57BL6J. Mice were housed under controlled lighting (12 h light/dark cycle) and temperature (22-24 °C) conditions. Food and water were available ad libitum.

**Surgical procedure**. Stereotaxic surgery was performed under anesthesia with a ketamine-xylazine mixture (100 and 10 mg/kg, respectively, i.p.) or with isoflurane using a vaporizer for small animals (Bio Research Center). For fiber photometric recordings, an optical fiber cannula (CFMC14L05, ∅ 400 μm, 0.39 NA; Thorlabs) was unilaterally implanted into the left primary motor cortex (+1.1 mm anteroposterior, +1.5 mm mediolateral from bregma, −1.2 mm dorsoventral from the skull surface, according to the Mouse Brain Atlas[68]). For local electrical stimulation, an optical fiber cannula attached to 50-μm diameter tungsten-wires (#2016971, California Fine Wire) was implanted. For virus injection, 33 gauge syringe needles (Hamilton) were used for infusion of AAV9-CaMKIIa-jGCaMP7f ($3.0 \times 10^{13}$ vg per mL) into the left primary motor cortex and a 0.2 μL viral solution was infused at a rate of 0.02 μL min$^{-1}$. For CBF measurements, the skull over the contralateral cortical hemisphere was exposed and sealed with silicone impression material (Shofu Inc.) until the beginning of the experiment. To fix the heads of mice, a U-shaped plastic plate was attached to the skull using dental cement (Fuji lute; GC Corporation) to enable its fixation to the stereotaxic frame during recordings.

For wide-field optical imaging, the skull over the entire cerebral hemispheres was exposed and sealed with an ultraviolet-curing jelly nail. To fix the head of mice under the microscope, a stainless chamber frame (Narishige) was attached to the skull using dental cement to enable its fixation to the stereotaxic frame during recordings.

Electrodes for EEGs and EMGs were implanted on the skull over the frontal cortex and neck muscles, respectively, with their reference electrode being implanted on the skull over the cerebellum. The mice were then housed separately for a recovery period of at least 5 days.

**Fiber photometry**. A fiber photometric system designed by Olympus Engineering (custom-made) was used to detect the compound intracellular ATP or calcium dynamics[19] (Fig. 2a). For ATP signal recordings, the input light (435 nm; silver-LED, Prizmatix) was reflected off a dichroic mirror (DM455CFP; Olympus), coupled into an optical fiber (M41L01, ∅ 600 μm, 0.48 NA; Thorlabs) linked to a next optical fiber (M79L01, ∅ 400 μm, 0.39 NA; Thorlabs) through a pinhole (∅ 600 μm), and then delivered to an optical fiber cannula (CFMC14L05; Thorlabs) implanted into the mouse brain. Light-emitting diode (LED) power was <200 μW at the fiber tip. Emitted yellow and cyan fluorescence light from ATeam was collected via an optical fiber cannula divided by a dichroic mirror (DM515YFP, Olympus) into cyan (referred as CFP; 483/32 nm band-pass filters, Semrock) and yellow (referred to as FRET; 542/27 nm band-pass filter; Semrock), and detected by two separate photomultiplier tubes (H7422-40; Hamamatsu Photonics). For Ca$^{2+}$ recording, input light (475 nm; silver-LED, Prizmatix) and dichroic mirrors (DM490GFP; Olympus and FF552-Di02-25×36; Semrock) were applied to detect GCaMP fluorescence (495-540 nm band-pass filter; Olympus). The fluorescence signals were digitized by a data acquisition module (NI USB-6008; National Instruments) and recorded by a custom-made LabVIEW program (National Instruments). Signals were collected at a sampling frequency of 1 kHz. The recordings were carried out under habituated head-fixed conditions in the latter half of the light phase.

**Wide-field optical imaging**. Wide-field imaging of ATP signals in vivo was performed using a macro-zoom fluorescence microscope (MVX-10, Olympus), with a high power LED (X-Cite 120LED, Lumen Dynamics Group Inc., wavelengths 385 nm and 430 nm) and an objective lens (1× MVX Plan Apochromat Lens, NA 0.25, Olympus). The microscope was equipped with a reverse dichroic mirror (U-MCFPHQ, Olympus), and the emission light was separated by a splitter (W-VIEW GEMINI, Hamamatsu Photonics) with band-pass filters (FF01-550/49-25 and FF01-496/20-25, Semrock) for the yellow and cyan light, respectively. For the recording of the entire hemisphere, a rectangular region (12 mm × 9 mm) was imaged at 320 × 240 pixels, at a focus set at 0.75 mm in depth from the top of the cortical surface. Images were acquired using the HCImage Live software at a frame rate of 1 Hz (exposure time: 500 ms) with a cooled CCD camera (ORCA-Flash4.0

V3, Hamamatsu Photonics). The recordings were carried out under habituated head-fixed conditions in the first half of the light phase.

**EEG, EMG, and CBF recordings**. EEGs and EMGs were always monitored during ATP measurements. During fiber photometric recordings, EEG and EMG signals were amplified (Model 3000, A-M systems), filtered, and digitized at 1 kHz using an analog-to-digital converter (USB-6008, National Instruments). EEG signals were high-pass- and low-pass-filtered at 0.1 Hz and 300 Hz, respectively. EMG signals were high-pass- and low-pass-filtered at 1 Hz and 300 Hz, respectively. The data acquisition software was written with LabVIEW (National Instruments). Simultaneously with the fiber photometric recordings, CBF was monitored using laser Doppler flowmetry (LDF; wavelength 780 nm; ATBF-LN1; Unique Medical) with a 0.5-mm needle probe attached to the skull over the contralateral cortical hemisphere, based on the CBF symmetry[69,70] (Fig. 3b). CBF signals were digitized at 1 kHz using an analog-to-digital converter (USB-6008).

During wide-field imaging, EEG and EMG signals were amplified (DAM50, WPI), filtered, and digitized at 1 kHz using an analog-to-digital converter (Micro1401-3, CED). EEG and EMG signals were high-pass- and low-pass-filtered at 0.1 Hz and 300 Hz, respectively. The data were recorded using the Spike2 software (CED).

**Cortical electrical stimulation**. A pulse generator with a constant current output (SEN-7203 and SS-202J, Nihon Kohden) delivered a square pulse (0.1-ms pulse width, 100 μA) at each of five frequencies (1 Hz for 5-s trains, 10, 30, 50, and 100 Hz for 3-s trains). Each animal received eight pulse trains at each frequency during the wake state, judged by online EEG/EMG recordings.

## Data analysis

*Vigilance state determination*. EEG/EMG recordings were automatically scored offline as wakefulness, NREM sleep, or REM sleep state using the SleepSign software version 3 (Kissei Comtec) in 4-s epochs according to standard criteria[71,72]. All vigilance state classifications assigned by SleepSign were examined visually and corrected if necessary. The same individual, blinded to genotype and the experimental condition, scored all EEG/EMG recordings. For detecting the onset of active-awake in the wake state and micro-awakening in the NREM sleep state, the time point at which EMG activity was above a threshold value (mean greater than 10 times the standard deviation) was obtained within each major state.

*Fiber photometric and CBF data analysis*. Under the electrical stimulation, ATP signals (a ratio of YFP (FRET) and CFP fluorescence intensity), CaMKII-GCaMP, and CBF signals were normalized between one (mean of 5 s before stimulation) and zero (trough value after 50 Hz stimulation) in each animal. Across the sleep-wake states, ATP signals were determined using a curve-fitting procedure and filtered with a Hamming window. CBF signals containing artifacts were removed and EMG signals were also filtered with a Hamming window. For the comparison of ATP/CBF levels across the states, the data immediately before and after the state transitions for each three consecutive epochs (12 s) were excluded from the analysis. Within each state, ATP/CBF signals (S) were normalized calculating the Z-score as $(S - S_{mean})/S_{SD}$, where $S_{mean}$ and $S_{SD}$ were the mean and standard deviation values of the wake state. Sleep-wake state transitions were defined as four consecutive epochs (16 s) of one state followed immediately by eight consecutive epochs (32 s) of a distinct state. Theoretically, six types of transitions could exist: wake-to-NREMS, wake-to-REMS, NREMS-to-wake, NREMS-to-REMS, REMS-to-wake, and REMS-to-NREMS. In practice, no animal displayed a 12-epoch sequence meeting the criteria for the REMS-to-NREMS or wake-to-REMS transitions, and thus data were only processed from the other four transition types. EEG delta power (1-4 Hz) was processed as raw values, and EEG theta frequencies (5-10 Hz) were divided by the delta frequencies for standardization. EMG activity was filtered with a Hamming window. Within the identified transitions, ATP and CBF signals, EEG parameters, and EMG activity were normalized calculating the Z-score as $(S - S_{mean})/S_{SD}$, where $S_{mean}$ and $S_{SD}$ were the mean and standard deviation values of the first three epochs of the wake-to-NREMS transition. All Z-scored data were averaged across all identified transitions exhibited by each animal. These average curves were subjected to two-way ANOVAs with epoch numbers as a within-subjects measure. For the onset of active-awake in the wake state and micro-awakening in the NREM sleep state, ATP/CBF signals and EMG activity averaged across all identified onsets exhibited by each animal were performed the adjustment of a zero point with the mean values of one epoch (4 s) immediately before the onset-time.

*Imaging data analysis*. Image analysis was performed using the ImageJ software (https://imagej.nih.gov/ij/). The original 512 × 512-pixel images were reduced to 64 × 64 pixels by binning. Regarding Fig. 4a, b, we calculated the Y/C ratio based on the complete 64 × 64 image dataset. The ATP signal was calculated by means of the fluorescence intensity of YFP (FRET) and CFP. Six circular ROIs were symmetrically placed across the entire cerebral cortex (Fig. 4c). Within each state, ATP signals were normalized calculating the Z-score as $(S - S_{mean})/S_{SD}$, where $S_{mean}$ and $S_{SD}$ were the mean and standard deviation values of the wake state to compare the ATP levels across the state. Sleep-wake state transitions were defined as four consecutive epochs (16 s) of one state followed immediately by eight consecutive

epochs (32 s) of a distinct state. Within the identified transitions, ATP and signals were normalized calculating the Z-score as $(S - S_{mean})/S_{SD}$, where $S_{mean}$ and $S_{SD}$ were the mean and standard deviation values of the first three epochs in a stage immediately before the transition. Cross-correlation was computed using the mean ATP signal of the ROIs. For the cross-correlation analysis, the data immediately before and after the state transitions for each 4 s were excluded from the analysis. All data analyses were performed using MATLAB (MathWorks).

**Statistics**. Statistics were calculated using MATLAB (MathWorks). The power spectral data of the Y/C ratios were obtained by a wavelet analysis. Two-sample comparisons were performed by paired t-test or Student's t-test. Multiple group comparisons were performed by repeated-measures ANOVA followed by Bonferroni post-hoc tests. The data with error bars are mean ± SEM in each graph, unless otherwise stated. All tests used are specified in the figure legends or text.

**Reporting summary**. Further information on research design is available in the Nature Research Reporting Summary linked to this article.

## Data availability
Source data used to generate the graphs in figures are provided in Supplementary Data 1. All data generated or analyzed during this study are available from the authors.

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

## Acknowledgements

This work was supported by a Grant for Research Fellow of the Japan Society for the Promotion of Science to A.N. (18K14756) and a Grant-in-Aids from Japan Foundation for Neuroscience and Mental Health, Narishige Neuroscience Research Foundation and Meiji Yasuda Life Foundation of Health and Welfare to A.N, PRESTO from JST (JPMJPR1887) to T.T, YoungGlia (Glial Assembly in Japan and Glial Heterogeneity in Germany) to A.N. and A.T. We also thank Seiichiro Sakai and Takashi Shichita for providing AAV9-CaMKIIa-jGCaMP7f. We would like to thank Editage (www.editage.jp) for English language editing.

## Author contributions

A.N. designed the research. A.N., T.T., and H.I. conducted the experiments. A.N., T.T, A.K., N.K., and T.S. analyzed the data. A.T., J.H., and K.N. established the Thy1-ATeam transgenic mouse line. T.K., K.M., N.T, K.M., K.F.T., and M.H. supervised the study. A.N. wrote the manuscript with input from all authors.

## Competing interests

The authors declare no competing interests.
