## [Peer Review File · Communications Biology]

Reviewers' comments:

Reviewer #1 (Remarks to the Author):

In the field of cellular energy homeostasis, several studies have indicated that mechanisms of neuronal energy homeostasis maintain a constant cellular energy value. In this report, the authors challenge this assumption, hypothesizing that global fluctuations in extracellular ATP levels of cortical neurons depend on sleep-wake states. To prove this hypothesis, they demonstrate that extracellular ATP levels in cortical excitation neurons fluctuate in a cortex-wide manner depending on the sleep-wake state.

The methods are sound, and the results clearly stated.

I find the paper interesting and worthy. Nonetheless, there are several points that I think the authors need to address before the manuscript is in an acceptable form.

1) Although I assumed the authors hypothesize that global fluctuations in extracellular ATP levels of cortical neurons depend on sleep-wake states, this is not explicitly stated. Maybe this is not the working hypothesis? Anyhow, it will strengthen the paper if they explicitly state their hypothesis.

2) They center their finding around the demonstration that ATP levels depend on sleep-wake states. They start their results, showing that ATP levels change with electrical stimulation. What was the aim of doing these experiments? The authors need to indicate the relevance of performing this part of the study. Besides, ATP homeostasis is robust under physiological conditions, but stimulating the cells for several seconds, beyond 20 HZ is not necessarily natural conditions. Cortical cells have a reported mean rate below 1Hz (Olshausen & Field, *Neural Computation* 17, 1665–1699 (2005)). It does not seem so surprising that with the intense electrical stimulation, ATP homeostasis breaks out of the tight control.

3) In the method sections, it is indicated that EEG/EMG recordings were scored by visual inspection. It is apparent that among the authors are experts in sleep research, but since the same people then compared ATP levels based on this characterization, there the risk of bias. This is particularly true for the characterization of microstates. Can instead, a simple automatic algorithm characterize the recording?

Reviewer #2 (Remarks to the Author):

The authors have attempted to address an important problem, the stasis and regulation on intracellular ATP levels in neurons in-vivo during various states of arousal or under field-electrode stimulation.

The greatest weakness and fundamental achille's heel with the work is the willingness to interpret the signals purely as an accurate reflection of ATP as opposed to something else. The principal confound in such measurements is likely to be pH. As metabolism is well known to drive changes in cytosolic proton concentration, changes for example in either glycolysis, lactate to pyruvate interconversion, and oxidative phosphorylation all are likely accompanied by changes in pH (and the Ca handling mechanisms in the cell do this as well). Although the original Imamura paper describing the FRET-based ATP reporter ATeam indicated that one of the constructs, AT1.03, likely was insensitive in the range of pH 7.3. Unfortunately the authors are not using this variant. Given that mVenus is famously pH sensitive and its FRET partner (CFP) much less so, changes in pH will almost certainly lead to changes in FRET. The authors need to provide convincing data that the signals they do measure are not in fact heavily corrupted by such artifacts. Unfortunately as the entire paper rests on assuming the

validity of the starting measurement one is left with little to be enthusiastic about.

We appreciate the critical review and insightful comments of both reviewers. Below, you will find our point-by-point responses to the reviewers. The reviewers' comments are numbered; our replies are in Gothic and revised sentences are in **Gothic**.

Reviewer #1:

In the field of cellular energy homeostasis, several studies have indicated that mechanisms of neuronal energy homeostasis maintain a constant cellular energy value. In this report, the authors challenge this assumption, hypothesizing that global fluctuations in extracellular ATP levels of cortical neurons depend on sleep-wake states. To prove this hypothesis, they demonstrate that extracellular ATP levels in cortical excitation neurons fluctuate in a cortex-wide manner depending on the sleep-wake state.

The methods are sound, and the results clearly stated.

I find the paper interesting and worthy. Nonetheless, there are several points that I think the authors need to address before the manuscript is in an acceptable form.

Response: We appreciate the reviewer's comments. It seems that there is a typographical error in the comments: intracellular ATP levels, instead of extracellular ATP levels. In this study, we focused on the dynamics of ATP levels in the cytoplasm of cortical neurons, whereas extracellular ATP fluctuations should also be extremely interesting.

1) Although I assumed the authors hypothesize that global fluctuations in extracellular ATP levels of cortical neurons depend on sleep-wake states, this is not explicitly stated. Maybe this is not the working hypothesis? Anyhow, it will strengthen the paper if they explicitly state their hypothesis.

Response: As Reviewer #1 indicated, the working hypothesis to examine the fluctuations in neuronal intracellular ATP levels across the sleep-wake states was not clearly stated in our original manuscript. When we started this study, we assumed that the neuronal intracellular ATP levels could be maintained constant across the sleep-wake states in the presence of brain energy homeostatic mechanisms. However, we observed the global state-dependent ATP dynamics in cortical neurons, which was beyond our expectation. We have stated our working hypothesis, with additional explanations regarding global brain energy homeostatic mechanisms, in the Introduction of the revised manuscript.

We have added the following sentences to the Introduction:

Introduction, Lines 55–65:

Furthermore, global brain energy homeostasis could also function in a state-dependent manner. Across the sleep-wake states of animals, brain metabolic activities for energy supply such as hemodynamics and glucose metabolism and multiple cellular energy-consuming activities including neuronal firings simultaneously fluctuate over wide brain areas⁵⁻⁸. In the wake state, brain-wide metabolic activities and neuronal activities increased, and in the non-REM sleep state, they decreased⁵⁻⁸. These global parallel dynamics of energy producing and consuming activities across the sleep-wake states could be brought about by the brain monoaminergic systems^{5,9}. Under these local and global brain energy homeostatic mechanisms, it is hypothesized that the cellular energy status in the brain could be maintained constant across the sleep-wake states of animals and cellular energy depletion could be prevented in all physiological conditions. However, this has not been experimentally proven.

Introduction, Lines 75–76:

To investigate whether the cellular energy status is maintained constant in the brain of living animals, we conducted in vivo optical measurements of intracellular ATP dynamics in neurons using a genetically encoded fluorescent ATP sensor^{11,14}.

2) They center their finding around the demonstration that ATP levels depend on sleep-wake states. They start their results, showing that ATP levels change with electrical stimulation. What was the aim of doing these experiments? The authors need to indicate the relevance of performing this part of the study. Besides, ATP homeostasis is robust under physiological conditions, but stimulating the cells for several seconds, beyond 20 HZ is not necessarily natural conditions. Cortical cells have a reported mean rate below 1Hz (Olshausen & Field, *Neural Computation* 17, 1665–1699 (2005)). It does not seem so surprising that with the intense electrical stimulation, ATP homeostasis breaks out of the tight control.

Response: We appreciate the reviewer's comment. In this study, we began with in vivo neuronal ATP measurements under local electrical stimulation, which is comparable with the previous ex vivo observation of neuronal ATP responses under electrical stimulation (Trevisiol A et al. (2017) eLife).

The aim of our in vivo electrical stimulation experiments was to evaluate the effect of local brain energy homeostatic functions to maintain the neuronal energy status constant, while stimulation with higher frequencies would induce the neuronal firings whose patterns were beyond the spontaneous ones (Olshausen & Field (2005) *Neural Comput.*). Note that our electrical stimulation protocols (stimulation intensity: 100 μ A and frequencies: 1–100 Hz) were adopted in reference to in vivo electrical stimulation protocols for inducing neurogenic vascular responses in previous studies (Iadecola C et al. (1997) *J Neurophysiol.*; Takano, T. et al. (2006) *Nat Neurosci.*), although these might induce beyond physiological conditions. We observed frequency-dependent decrease of neuronal ATP levels under electrical

stimulation whose conditions induced frequency-dependent hemodynamic responses in a parallel manner. These findings suggest that local brain energy homeostatic functions including hemodynamics could not be sufficient to always maintain neuronal ATP levels constant under the electrical stimulation, but could partly complement stimulation-induced neuronal energy consumption.

In addition, in vivo local electrical stimulation with various protocols have been classically adopted for neuronal activation in order to investigate neuronal function, such as neurotransmitter release and output behavior in animals (You ZB et al. (1998) J Neurosci.; Milad MR et al. (2004) Behav Neurosci.). Considering the classical use of in vivo electrical stimulation methodologies, our observation of electrical stimulation-induced long-lasting neuronal ATP reduction could be of interest.

We have added these explanations in the first paragraph of the Results section.

Results, Lines 93–96:

To evaluate the effect of local brain energy homeostatic functions to maintain the neuronal energy status constant, we observed in vivo neuronal intracellular ATP responses under local electrical stimulation, compatible with previous ex vivo observations¹¹.

Results, Lines 113–116:

Note that our electrical stimulation protocols (stimulation intensity: 100 μ A and frequencies: 1–100 Hz) were adopted in reference to in vivo electrical stimulation protocols for inducing neurogenic vascular responses in previous studies^{15,16}, whereas stimulation with higher frequencies would induce the neuronal firings whose patterns were beyond the spontaneous ones²⁴.

Results, Lines 134–139:

These findings suggest that local brain energy homeostatic functions, including hemodynamics, could not be sufficient to always maintain the neuronal ATP levels constant under the electrical stimulation but could slowly complement stimulation-induced neuronal energy consumption. Considering the classical use of in vivo electrical stimulation methodologies^{26,27}, our observation of electrical stimulation-induced long-lasting neuronal ATP reduction could be of interest.

3) In the method sections, it is indicated that EEG/EMG recordings were scored by visual inspection. It is apparent that among the authors are experts in sleep research, but since the same people then compared ATP levels based on this characterization, there the risk of bias. This is particularly true for the characterization of microstates. Can instead, a simple automatic algorithm characterize the recording?

Response: As we agreed with Reviewer #1's comment that visual inspection for scoring of states has a risk of bias, we automatically scored the EEG/EMG recording data as wakefulness, NREM sleep, or REM

sleep using the Sleepsign software (Kissei Comtec) according to standard criteria (Tobler et al. (1997) J Neurosci.; Yamanaka et al. (2002) Biochem Biophys Res Commun; Tsunematsu et al.(2014) J Neurosci.) in the revised manuscript.

Conversely, to detect substate changes within the major states (active-awake in wake state and micro-awakening in NREM sleep state), the time point at which EMG activity was above a threshold value (mean greater than 10 times the standard deviation) was automatically detected in each state. We had explained this in the “*Fiber photometric and CBF data analysis*” section in the Methods of the original manuscript; however, it might have easily been overlooked. In our revised manuscript, we have arranged this explanation in the “*Vigilance state determination*” section in the Methods (lines 437–440).

Based on the auto-scoring states, we re-analyzed and have replaced with all data of ATP and CBF dynamics in the Results section and in the figures (Figs. 2 to 3 and Figs. S3 to S5) of our revised manuscript. It is of note that our results and conclusion were not significantly altered by this change in analysis.

Before revision (Methods):

Data analysis

Vigilance state determination. EEG/EMG recordings were scored by visual inspection as wake, NREM sleep, or REM sleep in 4-s epochs using the following criteria: NREM sleep was characterized by high-amplitude EEG delta waves with low EMG power; REM sleep was characterized by low-amplitude, high-frequency EEG theta waves with very low EMG power; and wake was characterized by high and varying EMG power⁶⁴.

After (Methods, Lines 430–437):

Data analysis

Vigilance state determination. **EEG/EMG recordings were automatically scored offline as wakefulness, NREM sleep, or REM sleep state using the SleepSign software version 3 (Kissei Comtec) in 4-s epochs according to standard criteria^{71,72}. All vigilance state classifications assigned by SleepSign were examined visually and corrected if necessary.** The same individual, blinded to genotype and the experimental condition, scored all EEG/EMG recordings. **For detecting the onset of active-awake in the wake state and micro-awakening in the NREM sleep state, the time point at which EMG activity was above a threshold value (mean greater than 10 times the standard deviation) was obtained within each major state.**

Reviewer #2 (Remarks to the Author):

The authors have attempted to address an important problem, the stasis and regulation on intracellular ATP levels in neurons in-vivo during various states of arousal or under field-electrode stimulation.

The greatest weakness and fundamental achille's heel with the work is the willingness to interpret the signals purely as an accurate reflection of ATP as opposed to something else. The principal confound in such measurements is likely to be pH. As metabolism is well known to drive changes in cytosolic proton concentration, changes for example in either glycolysis, lactate to pyruvate interconversion, and oxidative phosphorylation all are likely accompanied by changes in pH (and the Ca handling mechanisms in the cell do this as well). Although the original Imamura paper describing the FRET-based ATP reporter ATeam indicated that one of the constructs, AT1.03, likely was insensitive in the range of pH 7.3. Unfortunately the authors are not using this variant. Given that mVenus is famously pH sensitive and its FRET partner (CFP) much less so, changes in pH will almost certainly lead to changes in FRET. The authors need to provide convincing data that the signals they do measure are not in fact heavily corrupted by such artifacts. Unfortunately as the entire paper rests on assuming the validity of the starting measurement one is left with little to be enthusiastic about.

Response: The reviewer's comment is truly insightful. To eliminate the most serious concern for interpretation of our data, we examined the pH dependence of the FRET signal of purified AT1.03^{YEMK} constructs, which is used in our study, by Imamura (Imamura et al. (2009) PNAS) (please see the below graph: Figure S1). The data showed that the fluorescence emission ratio of AT1.03^{YEMK} was almost invariant at pH conditions > 7.3. The ratio slightly decreased as the decrease of cellular pH lower than 7.3 when the cytosolic ATP levels were relatively high (higher than 2.0 mM). These findings suggest that the FRET signal from AT1.03^{YEMK} would not be heavily corrupted by pH fluctuations in cellular physiological conditions, i.e., approximately pH 7.3 and lower than 2 mM of neuronal [ATP] (Ainscow et al. (2002) J physiol.; Surin et al. (2014) Biochemistry; Rangaraju et al. (2014) Cell).

We have added the following data and explanations to our revised manuscript.

Figure S1. pH dependence of AT1.03^{YEMK}. The fluorescence ratios (527/475 nm) of purified AT1.03^{YEMK} constructs at 37°C at 0, 0.5, 1, 2, and 4 mM ATP in the pH range of 6.5–8.1 are shown. The buffer contained 50 mM Mops-KOH (pH 6.5–7.5) or Hepes-KOH (pH 7.7–8.1), 50 mM potassium chloride, 0.5 mM magnesium chloride, and 0.05% Triton X-100. Data are presented as mean ± SD.

Results, lines 104–109

As changes in pH can modulate fluorescence from some versions of YFP²⁰, we inspected the pH dependency of the FRET signal of purified AT1.03^{YEMK} constructs¹⁴. The fluorescence emission ratio was almost invariant at pH conditions > 7.3 (Fig. S1). The ratio slightly decreased as the decrease of cellular pH lower than 7.3 when the cytosolic ATP levels were higher than 2.0 mM. These findings suggest that the FRET signal from AT1.03^{YEMK} would not be heavily corrupted by cellular pH fluctuations²¹⁻²³.

We look forward to hearing from you regarding our submission. We would be glad to respond to any further questions and comments that you may have.

References

- Trevisiol, A. et al. Monitoring ATP dynamics in electrically active white matter tracts. *Elife* **17**, 6. pii: e24241 (2017).
- Olshausen, B.A. & Field, D.J. How close are we to understanding v1? *Neural. Comput.* **8**, 1665-1699 (2005).
- Iadecola, C., Yang, G., Ebner, T.J. & Chen, G. Local and propagated vascular responses evoked by focal synaptic activity in cerebellar cortex. *J. Neurophysiol.* **78**, 651-659 (1997).
- Takano, T. et al. Astrocyte-mediated control of cerebral blood flow. *Nat. Neurosci.* **9**, 260-267 (2006).
- You, Z.B., Tzschentke, T.M. & Brodin, E., Wise, R.A. Electrical stimulation of the prefrontal cortex increases cholecystokinin, glutamate, and dopamine release in the nucleus accumbens: an in vivo microdialysis study in freely moving rats. *J. Neurosci.* **18**, 6492-6500 (1998).
- Milad M.R., Vidal-Gonzalez, I. & Quirk, G.J. Electrical stimulation of medial prefrontal cortex reduces conditioned fear in a temporally specific manner. *Behav. Neurosci.* **118**, 389-394 (2004).
- Tobler, I., Deboer, T. & Fischer, M. Sleep and sleep regulation in normal and prion protein-deficient mice. *J. Neurosci.* **17**, 1869-1879 (1997).
- Yamanaka, A., et al. Orexins activate histaminergic neurons via the orexin 2 receptor. *Biochem. Biophys. Res. Commun.* **290**, 1237–1245 (2002).
- Tsunematsu, T, et al., Optogenetic manipulation of activity and temporally controlled cell-specific ablation reveal a role for MCH neurons in sleep/wake regulation. *J. Neurosci.* **34**, 6894-6909 (2014).
- Imamura, H. et al. Visualization of ATP levels inside single living cells with fluorescence resonance energy transfer-based genetically encoded indicators. *Proc. Natl. Acad. Sci. USA.* **106**, 15651-15656 (2009).

- Ainscow, E.K. et al. Dynamic imaging of free cytosolic ATP concentration during fuel sensing by rat hypothalamic neurones: evidence for ATP-independent control of ATP-sensitive K(+) channels. *J. Physiol.* **544**, 429-445 (2002).
- Surin, A.M. et al. Study on ATP concentration changes in cytosol of individual cultured neurons during glutamate-induced deregulation of calcium homeostasis. *Biochemistry (Mosc)*. **79**, 146-157 (2014).
- Rangaraju, V., Calloway, N. & Ryan, T.A. Activity-driven local ATP synthesis is required for synaptic function. *Cell* **156**, 825-835 (2014).

REVIEWERS' COMMENTS:

Reviewer #4 (Remarks to the Author):

I have read both the pH independence concern of reviewer #2 and the authors' response.

My take is the following:

To address the concern raised by Reviewer #2 - that the AT1.03YEMK version might show a high pH dependence undermining its ATP reporting value - the authors opted to perform a pH-dependence analysis of AT1.03YEMK. Specifically, the fluorescence ratios (527/475 nm) at 37 °C at 0, 0.5, 1, 2, and 4 mM ATP in the pH range of 6.5–8.1 were analysed (Figure S1). The experimental approach is comparable to the one performed by Imamura et al. (2009) in PNAS for the AT1.03 version (Figure 3B).

The AT1.03YEMK version appears not to be affected by small pH fluctuations around normal cytoplasmic pH values, which is near 7.3. Its pH-independence is slightly inferior to the AT1.03 version, which seems slightly less affected at values of around pH 7.1. This difference does not seem to be of any major concern though. The response is therefore adequate.